# Polypropylene Graft Poly(methyl methacrylate) Graft Poly(*N*-vinylimidazole) as a Smart Material for pH-Controlled Drug Delivery

**DOI:** 10.3390/ijms23010304

**Published:** 2021-12-28

**Authors:** Felipe López-Saucedo, Jesús Eduardo López-Barriguete, Guadalupe Gabriel Flores-Rojas, Sharemy Gómez-Dorantes, Emilio Bucio

**Affiliations:** Departamento de Química de Radiaciones y Radioquímica, Instituto de Ciencias Nucleares, Universidad Nacional Autónoma de México, Circuito Exterior, Ciudad Universitaria, Mexico City 04510, Mexico; jelbarrig@gmail.com (J.E.L.-B.); ggabofo@hotmail.com (G.G.F.-R.); shar.hj@gmail.com (S.G.-D.)

**Keywords:** grafting, polypropylene, gamma rays, methyl methacrylate, *N*-vinylimidazole, pH-responsiveness, vancomycin, release

## Abstract

Surface modification of polypropylene (PP) films was achieved using gamma-irradiation-induced grafting to provide an adequate surface capable of carrying glycopeptide antibiotics. The copolymer was obtained following a versatile two-step route; pristine PP was exposed to gamma rays and grafted with methyl methacrylate (MMA), and afterward, the film was grafted with *N*-vinylimidazole (NVI) by simultaneous irradiation. Characterization included Fourier transform infrared spectroscopy (FTIR), scanning electron microscope (SEM), thermogravimetric analysis (TGA), X-ray photoelectron spectroscopy (XPS), and physicochemical analysis of swelling and contact angle. The new material (PP-*g*-MMA)-*g*-NVI was loaded with vancomycin to quantify the release by UV-vis spectrophotometry at different pH. The surface of (PP-*g*-MMA)-*g*-NVI exhibited pH-responsiveness and moderate hydrophilicity, which are suitable properties for controlled drug release.

## 1. Introduction

The surface functionalization of polypropylene using conventional chemical methods presents well-known difficulties [1]. Such difficulties are attributed to the thermal stability and lack of reactivity from alkyl chains in the polymer. Nonetheless, the modification of polymer materials is often a prolific rewarding task because the addition of functional groups, whether in bulk or only in a specific part [2], is a standardized strategy to provide new physical and chemical properties to materials to boost their functionality [3]. 

Grafting polymerization has been demonstrated as an excellent option to modify PP [4], where different methods [5] have been tried to graft vinyl monomers, such as *N*-vinylimidazole (NVI) [6], methyl methacrylate (MMA) [7], *N*-vinylcaprolactam (NVCL) [8], or glycidyl methacrylate (GMA) [9], among others. However, there are issues related to the low reactivity of PP, which produces non-uniform grafting, low yields, waste, and residues [10].

Currently, alternative energy sources are becoming more relevant to carry out grafting polymerization, such as gamma rays [11], plasma [12], UV-light, and electron-beam [13]. Among these energy sources, high energy gamma rays of 1.17 and 1.33 MeV from ^60^Co [14] are suitable to promote the homolytic rupture of stable C-H and C-C bonds from PP polymeric chains. Exposing PP to gamma rays produces free radicals (unstable), which are stabilized as peroxides and hydroperoxides under an oxidizing atmosphere [15]. The said process is named the “pre-irradiation oxidative method” and is used to induce grafting polymerization onto PP surfaces with vinyl monomers [2]. The grafting degree of vinyl monomers depends on different factors, such as the solvent, time, reaction temperature, and absorbed dose. In general terms, a good understanding of reactants ensures obtaining materials with the desired properties. 

Therefore, gamma-ray-induced graft polymerization can provide new materials that meet multifunctional or multipurpose needs [3]. MMA is a methacrylic monomer used for multiple purposes with an aliphatic part (non-polar) and a carbonyl group (polar) [16]. In addition, NVI is a vinyl molecule with an N heteroatom ring used for diverse objectives [17,18]. Both carbonyl and imidazole groups can be employed in drug delivery systems, thanks to electrostatic and hydrophobic interactions drug-polymer. Hence, these polymers, PP, PMMA, and PNVI, may be implemented to design new materials with tailored properties for biomedical devices considering their biocompatibility [12,19,20].

In summary, this work presents grafting polymerization of MMA and NVI onto PP with the subsequent loading and release of vancomycin [21], where the pH-responsiveness of NVI chains [22] was studied as well as other physicochemical properties, such as swelling and contact angle.

## 2. Results

The materials were modified successfully with MMA by grafting polymerization using the pre-irradiation oxidative method. The grafting degree showed a dependence on the absorbed dose, temperature, monomer concentration, and reaction time, offering an excellent control on the yield and leading to the possibility of obtaining tailored grafted PP films. In the case of the grafting degree of NVI, this was carried out by the direct method and did not show a considerable grafting yield if compared to MMA graft, which was more quantitative.

### 2.1. Grafting

During the grafting process on PP films, it is possible to observe certain tendencies regarding the grafted acrylate. Grafting of MMA exhibited a linear slope in the absorbed dose experiment (5 to 25 kGy) and in the time reaction experiment (5 to 26 h) reaching a maximum grafting of 49.5% (25 kGy and 16 h) and 31% (5 kGy and 26 h), respectively (Figure 1a,b). However, the absorbed dose of 25 kGy could cause a detriment or deterioration on the PP matrix caused by polymer chain rupture, cross-linking, and increment of oxygenated groups [23]; for this reason, 5 kGy is the absorbed dose preferred.

Graft by varying temperature and monomer concentration completed the grafting study. Regarding the effect of monomer concentration, it had a maximum grafting of 77.5% with a linear tendency (Figure 1c), but at the lower monomer concentrations (20%), the graft was adequate to incorporate a superficial modification. Finally, the results indicated that the minimum activation temperature for this system is about 50 °C, which is congruent with the temperature to activate peroxides. The grafting degree increased progressively up to 80 °C, but at 90 °C the graft slightly decreased, indicating that when the reaction took place at 90 °C, the homopolymerization was benefited, so the graft was affected (Figure 1d). Therefore, the results suggest a possible control on the grafting rate either by reaction time or reaction temperature, offering a reasonable percentage of functionalization using a low absorbed dose of 5 kGy and low monomer concentration, thus, ensuring lower damage in the properties of the matrix.

The grafting of NVI was carried out on PP-*g*-MMA with different grafting degrees from 8.5 to 77.5%. The NVI grafting degree was lower compared to the results obtained in the MMA grafting, which indicates a lower monomer reactivity. Since the MMA grafting degree of PP-*g*-MMA was higher, the grafting yield of NVI did not increase proportionally, obtaining yields ranging from 4 to 6.5%. Hence, it is understood that a slight modification with MMA is enough to promote the graft of NVI in a second step (Figure 2). In the following lines, the notation (PP-*g*-MMA)-*g*-NVI (x/y%) represents the binary grafted weight percent of “x” PMMA and “y” PNVI, respectively. 

SEM microscopy was performed to analyze the surface morphology of PP-*g*-MMA (17%) and (PP-*g*-MMA)-*g*-NVI (19.5/6%) (Figure 3). Morphological changes due to the grafting polymerizations were observed, clearly indicating a surface copolymerization with an amorphous appearance, which is suitable for the adsorption of solids, as was found in this case.

### 2.2. Infrared Spectroscopy

For pristine PP, as a linear polymer constituted just of propylene units, strong bands of infrared corresponding to different modes of C-H vibration were displayed, which stretch in the region of 2949–2838 cm^−1^ and bend methyl (-CH_3_) and methylene (-CH_2_) groups at 1456 and 1375 cm^−1^, respectively (Figure 4). Once the first copolymer was achieved, the spectrum of PP-*g*-MMA, besides the aliphatic bands, showed the characteristic carbonyl band around 1724 cm^−1^, which appeared as a strong signal accompanied by the C-O stretching at 1145 and 1063 cm^−1^ [24]. After the second graft with NVI [25], in addition to the mentioned bands, there was an aromatic C-H stretching band at 3112 cm^−1^ and the characteristic bands of aromatic compounds in the fingerprint region between 900 and 650 cm^−1^ [26,27].

### 2.3. XPS Spectroscopy

XPS study determined the surface atomic compositions of pristine PP [28] and grafted films, confirming the existence of grafted PMMA [29] and PNVI [27] on the surface, as shown in Figure 5. The characteristic peaks of carbon (C 1 s at 285.0 eV), oxygen (O 1 s at 531.0 eV), and nitrogen (N 1 s at 399.4 eV) were detected in the scanning, and the atomic level relationship was obtained from the core level peak areas of C 1 s, O 1 s, and N 1 s, and multiplied by the corresponding sensitivity factors giving the results in Table 1.

### 2.4. Thermal Gravimetry Analysis

Thermograms of grafted films displayed a faster weight loss compared to the TGA curve observed in the pristine non-irradiated PP film, as the 10% weight loss indicates (Figure 6). Decomposition temperature (Td) of pristine PP was higher than in the PP-*g*-MMA (25%) and (PP-*g*-MMA)-*g*-NVI (19.5/6%). The grafted films exhibited a multi-step decomposition, as is shown in the thermogram of PP-*g*-MMA (25%), where there were two decomposition stages, while in the (PP-*g*-MMA)-*g*-NVI (19.5/6%), there were three decomposition stages (Table 2). In conclusion, the study showed that pristine PP had better thermal stability in comparison to grafted films, but this difference is merely informative because grafted films worked well at load and release temperatures.

Multiple decomposition stages in the grafted films suggest a localized polymer composition forming a multilayer material. These zones are core, internal layer, and surface; nonetheless, the PP zone is in the nucleus and preserves its inherent thermal properties. This characteristic is found in a surface-grafting polymer [30].

### 2.5. Swelling and Critical pH 

The unmodified and grafted PP films were put to swelling tests by immersion in different solvents for 24 h to determine their behavior in liquid mediums. Solvents were chosen according to their dielectric constant (ε), in order of polarity: water (78.5), dimethyl formaldehyde (DMF) (38.25), methanol (32.6), n-propanol (20.1), toluene (2.38), and n-hexane (1.89). The non-polar solvents n-hexane and toluene swelled the films more than the other solvents. One parameter for choosing a suitable solvent to graft NVI is its capability of swelling the film PP-*g*-MMA and as was expected, the highest swellings were achieved in toluene because both PP-*g*-MMA and (PP-*g*-MMA)-*g*-NVI have non-polar groups in their chains. These preliminary tests also helped to determine the viability of water for the load/release assays (Figure 7), although the water had the lowest swelling percentage followed by DMF, methanol, and n-propanol. 

One of the characteristics of NVI-containing polymers is their pH response. This property is conferred by grafted PNVI, an electron donor polyelectrolyte (i.e., a Lewis base) that shrinks or expands by varying pH, as was verified with the study of swelling in a pH range of 3 to 11. In this case, two films of (PP-*g*-MMA)-*g*-NVI with different compositions were analyzed, as shown in Figure 8. The behavior was similar in both films since they were hydrophilic at acid pH and hydrophobic at alkaline pH, with inflection points at pH 6.9 for (PP-*g*-MMA)-*g*-NVI (20/6%) and pH 7.3 for (PP-*g*-MMA)-*g*-NVI (34/6.5%), respectively. The most significant difference was the higher swelling in the film with more PMMA grafted, so it is inferred that this polyacrylate conferred a more hydrophilic behavior to the film. Although swelling between 1 and 4% would seem low, the thickness of the films was 18 mm, and samples were above 250 mg, so even small weight changes, such as 0.1 mg, were detected. Furthermore, the swelling percentage was enough to load and release the vancomycin quantitatively (see Section 2.7), because in this case, the swelling occurred exclusively on the surface. Thus, it is concluded that the surface of the grafted films is moderately hydrophilic and pH-responsive can uptake water and molecules between their chains.

### 2.6. Contact Angle

Once MMA is grafted onto the surface, the wettability was an important parameter because the acrylic chains are moderately hydrophilic, as swelling experiments showed (see Section 2.5). Thereby, the contact angle before and after graft can be interpreted in terms of surface energy, increasing in this case, which means that grafted films were more hydrophilic than the pristine PP films, even when the grafted chains had a hydrophobic moiety, the amorphous acrylic chains increased the wettability (Figure 9). The contact angles determined by the drop of water on the pristine PP film were 87.5 ± 0.9° at 1 min and 84.4 ± 0.9° at 5 min, and the highest compared with the grafted films, in which case the contact angle decreased as the MMA graft percentage increased. The change from hydrophobic to hydrophilic occurred since the first sample with the lowest graft that was PP-*g*-MMA (10%), with angles of 76 ± 1.7° at 1 min and 71.2 ± 2.3° at 5 min and decreased consecutively up to the last sample, PP-*g*-MMA (50%), exhibiting a contact angle of 64.5 ± 4.4° at 1 min and 59.4 ± 4.8° at 5 min, being the lowest angle and therefore the most hydrophilic.

### 2.7. Load and Release of Vancomycin

Drug loading was performed using the sample (PP-*g*-MMA)-*g*-NVI (23/5.5%) in an aqueous solution of vancomycin hydrochloride [2 mg mL^−1^]. The main reasons for choosing water as solvent were its capability to dissolve the drug and its innocuousness. These properties are more relevant than the limited capability to swell the film [31].

Once the vancomycin was loaded, the release was in a controlled pH [32], in both acid (pH 4–6) and alkaline (pH 8–9) buffer medium to determine the release rate in simulated physiological conditions, considering the pH responsiveness of PNVI grafted [33], particularly at a pH close to that of the skin [34,35].

At neutral pH, the release rate was the highest, and the maximum reached within the first 2 h, which was 109.5 ± 4.3 μg cm^−2^ (Figure 10a), which was around 83% of the total loaded vancomycin. While in alkaline and acid pH, the release rate and the amount of vancomycin released decreased considerably (Figure 10b). It was found that even after 48 h, the release value at pH 8 was only 47.1 ± 2.5 μg cm^−2^. In all buffers, unlike neutral pH, release rates were slower, and the maximum concentration was not reached at 48 h. When the kinetics were checked, it was found that they followed a different path than at neutral pH, given that under neutral conditions the release rate was around three times and in a fraction of the time (2 h). Therefore, there is an expedited diffusion at pH 7 and a prolonged release at pH 4, 5, 8, and 9.

## 3. Discussion

The difficulty of grafting NVI directly onto pristine PP was overcome by the easiness of grafting MMA. The difference among the reactivity of MMA and NVI is attributed to electronic effects, solvent, stability of intermediates (during chain reaction), and homopolymerization rate (reaction in competition during the copolymerization). The surface of PP grafted films exhibited changes in their hydrophilic/hydrophobic behavior and became able to load and release vancomycin in different pH conditions. These grafted materials are not limited to the vancomycin since the drug loading by swelling is a general method for delivery with many bioactive principles [36].

SEM analysis suggested that the graft of both copolymers took place on the surface of PP-*g*-MMA and (PP-*g*-MMA)-*g*-NVI films, which was supported by observing amorphous layers and by the fact that there was no significant change in the film’s size. This information was consistent with infrared, where it was possible to observe the corresponding bands of the acrylic chains on the surface. While in XPS, elemental analysis detected the presence of O and N from the graft. Finally, TGA confirmed that grafted PP was thermally stable [37], even when the amorphous grafted binary and single copolymers decomposed earlier than pristine PP.

Regarding the swelling properties, grafted films showed a slight swelling on the polar solvents, which is a significant difference with pristine PP because the polar solvent absorbs between the alkyl backbones. The water drop contact angle showed that the surface of the PP-*g*-MMA was wet consistently as the contact angle decreased while the grafting degree increased. Overall, the swelling and contact angle results suggest an increase in the surface energy of the grafted films and are suitable to use in drug release systems [38]. 

The release rate of vancomycin on (PP-*g*-MMA)-*g*-NVI (23/5.5%) was studied, finding that this molecule, with several amines and one carboxylic group, at an acid or alkaline pH formed strong H interactions with the grafted chains, prolonging the release and decreasing the release rate [39]. At these conditions, the release could be conducted by a simultaneous equilibrium, with an interchange of ions from the system and medium [40]. However, the critical pH was reached at neutral conditions, where there was no excess of H^+^ or OH^−^ ions, which eased the release due to the strong interaction among medium and grafted chains [41,42], yielding 83% of the total drug released in the first hours.

## 4. Materials and Methods

### 4.1. Materials

High-density polypropylene films (0.18 cm thickness) were from Goodfellow (Huntingdon, Cambridgeshire, UK). Vancomycin hydrochloride, MMA (99%), and NVI (99%) were purchased from Aldrich Chemical Co. (Saint. Louis, Missouri, USA), and monomers were purified by vacuum distillation. Boric acid, citric acid, trisodium orthophosphate, and solvents (including double distilled water) were acquired from Baker (Mexico City, Mexico).

### 4.2. Grafting Method

MMA was grafted by oxidative pre-irradiation method, and NVI was grafted by direct method [43], in both cases using a gamma-rays source of ^60^Co Gammabeam 651-PT (UNAM, Mexico City, Mexico) at a dose rate of 8.4 kGy h^−1^, the methods are described in detail in the next Section 4.2.1 and Section 4.2.2.

#### 4.2.1. Grafting Polymerization of MMA Using the Oxidative Pre-Irradiation Method

PP films of 3cm × 2cm × 0.18 cm (width, length, and thickness) were weighed (around 250 mg) and placed into open glass ampoules and exposed to gamma irradiation in the presence of air. Afterward, the solutions of MMA were prepared in methanol as the solvent at different concentrations (Table 3) and then added (5 mL) into the glass ampoules with the pre-irradiated PP film. The ampoules were degassed by freezing and thawing cycles with liquid nitrogen followed by a purged in the vacuum line; subsequently, the ampoules were sealed at vacuum. Then, the polymerization was initiated by heating in a water bath at different times and temperatures. Once completed the reaction time, the ampoules were open, and the films were rinsed in a water/ethanol mixture 50/50 vol% under constant stirring for 24 h. Finally, the samples were dried in a vacuum oven at 60 °C for 24 h. The grafting percentage was calculated according to Equation (1), using the weight of pristine (W_0_) and grafted (W_g_) film.
Grafting (%) = 100[(W_g_ − W_0_)/W_0_](1)

#### 4.2.2. Grafting of NVI on PP-*g*-MMA by Direct Method 

PP-*g*-MMA films with different MMA grafting degree (Table 4) were weighed and placed into glass ampoules containing 6 mL of NVI in toluene (50 vol%). Then, oxygen was removed from the ampoules with freezing and thawing cycles (see Section 4.2.1), then the ampoules were sealed with a blowtorch and irradiated with an absorbed dose of 15 kGy at room temperature. Finally, the ampoules were open, and the grafted films were rinsed with methanol and dried into a vacuum oven at 60 °C for 24 h. The weight of films was recorded to calculate the grafting degree according to Equation (1).

### 4.3. Swelling Experiments

The samples were placed in different solvents until they reached the limit swelling (maximum 24 h) at room temperature (around 25 °C). Excess solvent was removed with an absorbent paper. The solvents used for swelling were water, methanol, n-propanol, DMF, and n-hexane. The swelling percentage (%) was calculated according to Equation (2):Swelling (%) = 100[(W_s_ − W_d_)/W_d_](2)
where W_s_ and W_d_ are the weights of swollen and dried films, respectively.

The (PP-*g*-MMA)-*g*-NVI (23/5.5%) film was employed to determine the critical pH. The sample was put inside different phosphate buffers (pH 4–9) for 24 h to record the weight. After each measurement, the sample was rinsed with double distilled water and submerged in the next buffer. Equation (2) was also applied to calculate the swelling at different pH.

### 4.4. Load and Release of Vancomycin

Vancomycin was loaded to the (PP-*g*-MMA)-*g*-NVI (23/5.5%) film. A fresh dissolution of vancomycin hydrochloride [2 mg mL^−1^] was prepared with double distilled water and poured into a vial containing the grafted film. The vial was stored in refrigeration at 4 °C for 48 h; then, the film was taken out, dried, and stored at room temperature (around 25 °C). The amount of vancomycin loaded was calculated by measuring the vancomycin released under sonication and replacing the solvent (double distilled water) until reaching absorbances close to 0, these absorbances represent the total vancomycin concentration, resulting 132.2 ± 0.8 μg cm^−2^.

Release experiments were performed using the same film which was (PP-*g*-MMA)-*g*-NVI (23/5.5%) in 4 mL of sodium phosphate buffer (pH 4–9), 0.1 M, and at 37 °C. The releasing was monitored at different times, recording absorbances by spectrophotometry at 280 nm [44].

### 4.5. Instrumental

Infrared spectroscopy attenuated total reflection (FTIR-ATR) spectra of dry pristine and modified films were analyzed using a Perkin–Elmer Spectrum 100 spectrometer (Norwalk, CT, USA) with 16 scans.

X-ray photoelectron spectroscopy was performed in an ultra-high vacuum (UHV) system Scanning XPS microprobe PHI 5000 Versa Probe II (Chanhassen, MN, USA), with an excitation source of Al Kα monochromatic, energy 1486.6 eV, 100 µm beam diameter, and with a Multi-Channel Detector (MCD). The XPS spectra were obtained at 45° to the normal surface in pass energy mode (CAE) E0 = 117.40 and 11.75 eV. Peak positions were calibrated to Ag 3d5/2 photopeak at 368.20 eV, having a full width at half maximum of 0.56 eV, and the energy scale corrected using the C 1s peak brought to 285.0 eV.

A Kruss DSA 100 drop shape analyzer (Matthews, NC, USA) was employed to measure water droplet contact angle at 1 and 5 min in triplicates.

Scanning electron microscope (SEM) images were acquired by the Zeiss Evo LS15 instrument (Jena, Germany). Small pieces (1 cm length) of grafted samples were cut and directly analyzed under a high vacuum without using any coating.

Thermogravimetric analysis (TGA) data of weight loss and decomposition of pristine and modified films (around 10 mg) were analyzed under a heating rate of 10 °C min^−1^ and run from 20 to 800 °C in a TGA instrument Q50 TA Instruments (New Castle, DE, USA).

Ultraviolet-visible (UV-vis) spectrophotometer model Agilent 8453 (Waldbronn, Germany) was utilized to analyze the release of vancomycin at 280 nm, using quartz cuvettes (1 cm length).

## 5. Conclusions

Radiation-grafting was a convenient method to modify the surface of PP films. In the first step, the PP-*g*-MMA was obtained by the pre-irradiation oxidative method, and in the second step, the final material (PP-*g*-MMA)-*g*-NVI was achieved by simultaneous irradiation. The grafting of NVI endowed the surface with pH responsiveness and the chains were able to load vancomycin hydrochloride in aqueous dissolution (2 mg mL^−1^). The release of vancomycin was pH dependent with a higher rate at pH 7 and more controlled release at non-neutral pH; the maximum amount of drug released at buffer pH 7 was 109.5 ± 4.3 μg cm^−2^ after 48 h. These findings suggest an active interaction in the equilibrium of the NVI chains-vancomycin-release medium. This type of superficial modification onto a non-reactive thermoplastic, such as the PP, provides a route to get more sophisticated materials and devices.

## Figures and Tables

**Figure 1 ijms-23-00304-f001:**
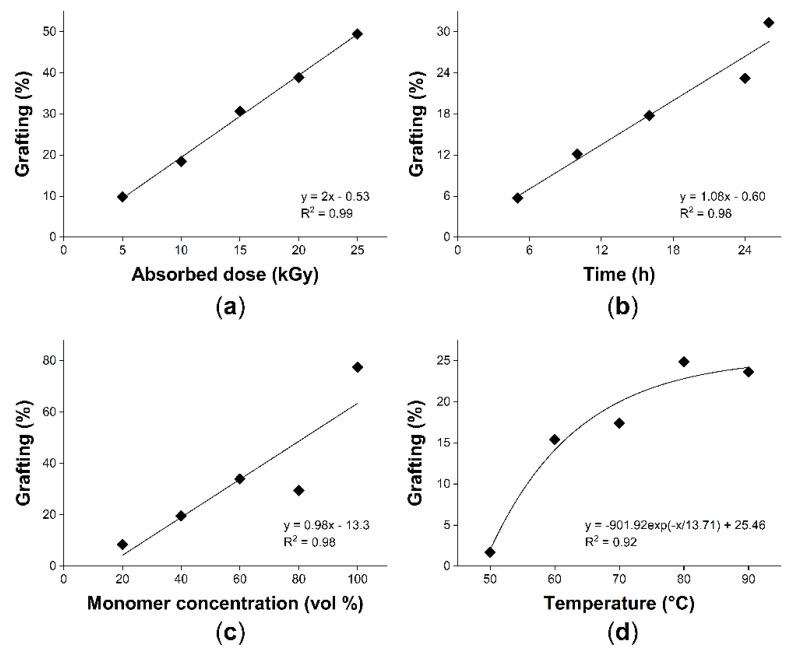
Grafting of MMA onto pristine PP: (**a**) Effect of absorbed dose (16 h, 65 °C, MMA 20 vol%); (**b**) reaction time (5 kGy, 65 °C, MMA 30 vol%); (**c**) monomer concentration (15 kGy, 16 h, 70 °C), and (**d**) reaction temperature (5 kGy, 16 h, MMA 30 vol%).

**Figure 2 ijms-23-00304-f002:**
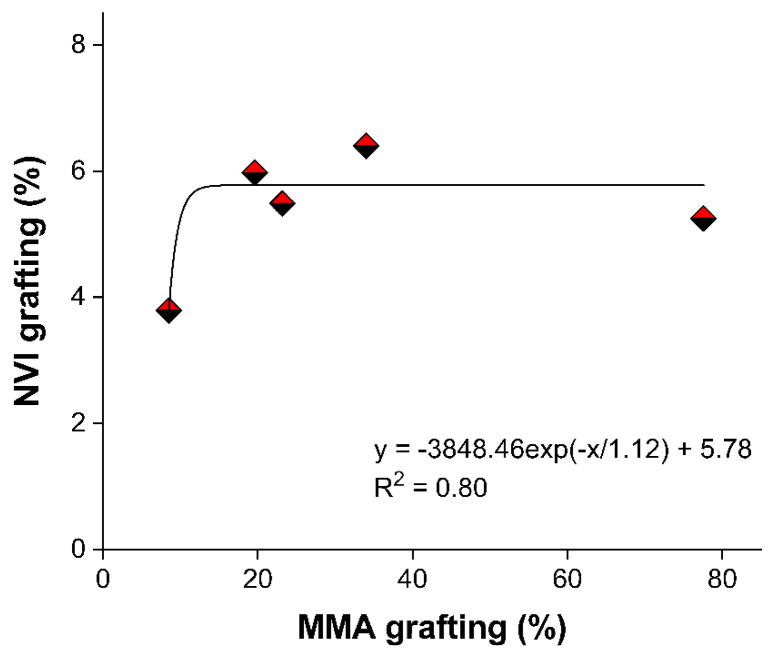
Results of NVI grafted on PP-*g*-MMA (50%); reaction conditions, absorbed dose 15 kGy, and room temperature (around 25 °C).

**Figure 3 ijms-23-00304-f003:**
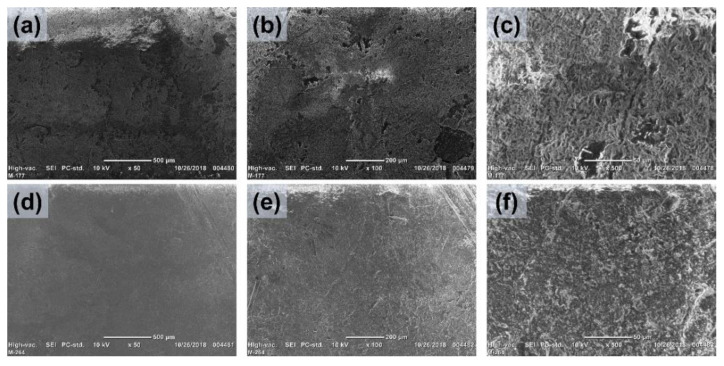
SEM images of (**a**–**c**) PP-*g*-MMA (17%) and (**d**–**f**) (PP-*g*-MMA)-*g*-NVI (19.5/6%), augmented from left to right ×50, ×100, and ×500.

**Figure 4 ijms-23-00304-f004:**
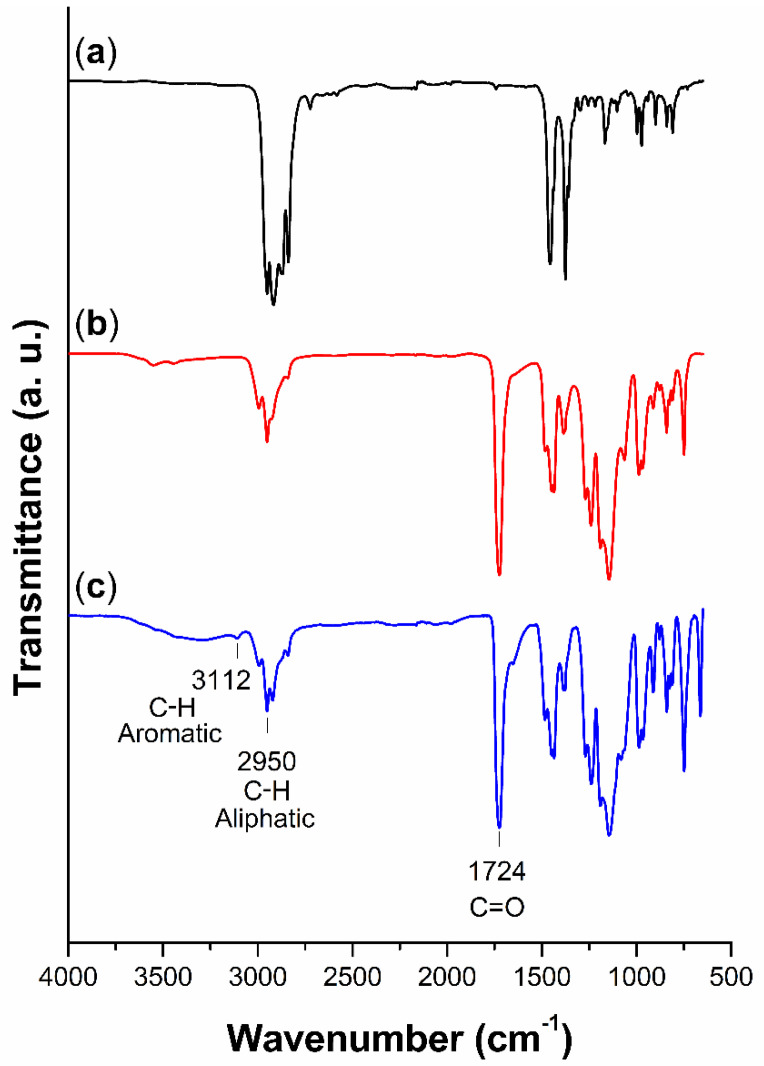
FTIR spectra: (**a**) non-irradiated PP, (**b**) PP-*g*-MMA (10%), and (**c**) (PP-*g*-MMA)-*g*-NVI (77.5/5%).

**Figure 5 ijms-23-00304-f005:**
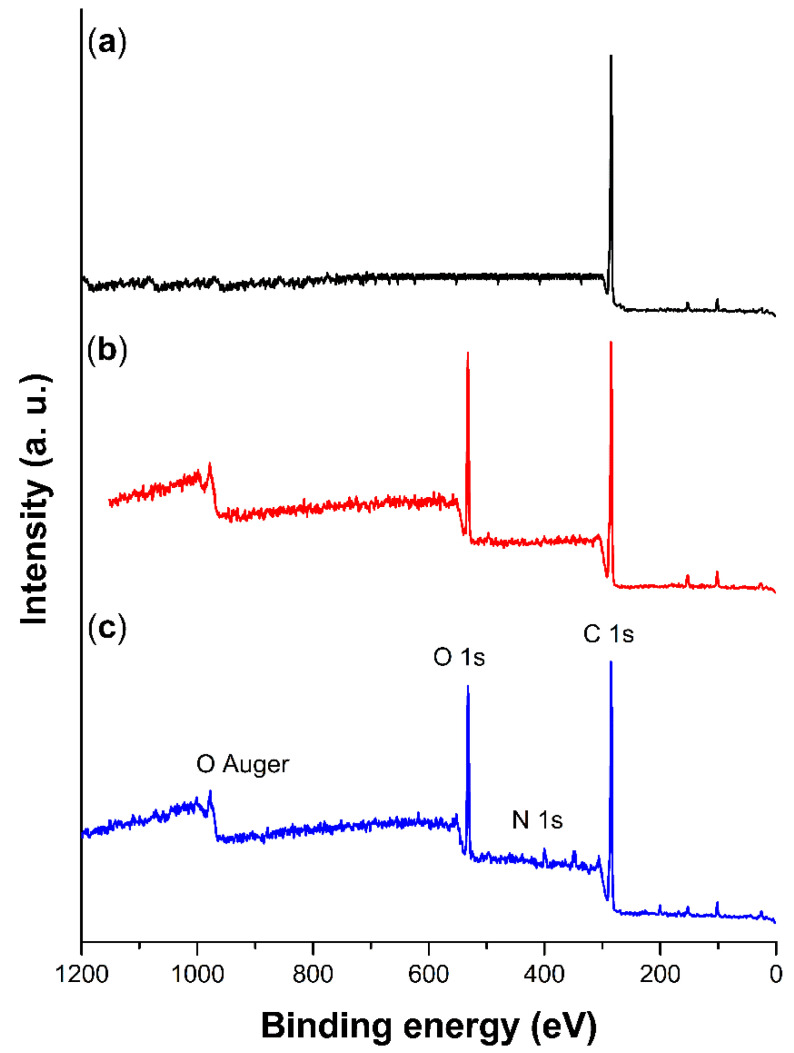
XPS scan of: (**a**) non-irradiated PP, (**b**) PP-*g*-MMA (17%), and (**c**) (PP-*g*-MMA)-*g*-NVI (77.5/5%).

**Figure 6 ijms-23-00304-f006:**
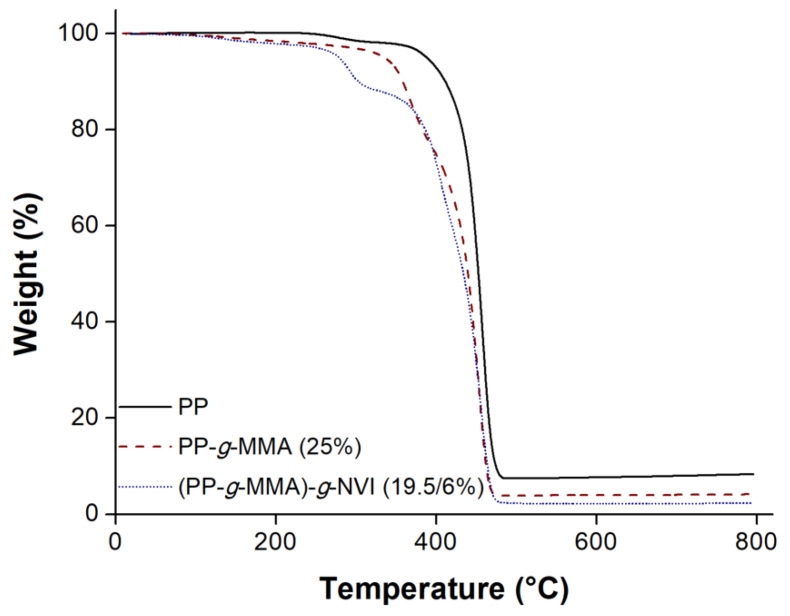
Thermogram runs under nitrogen atmosphere at 800 °C and heating rate 10 °C min^−1^.

**Figure 7 ijms-23-00304-f007:**
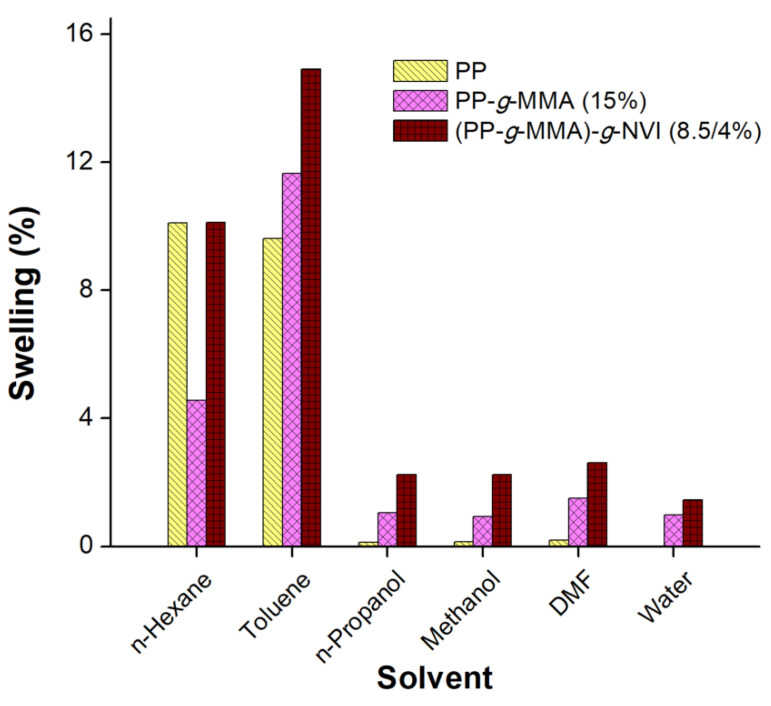
Swelling of the films in different solvents.

**Figure 8 ijms-23-00304-f008:**
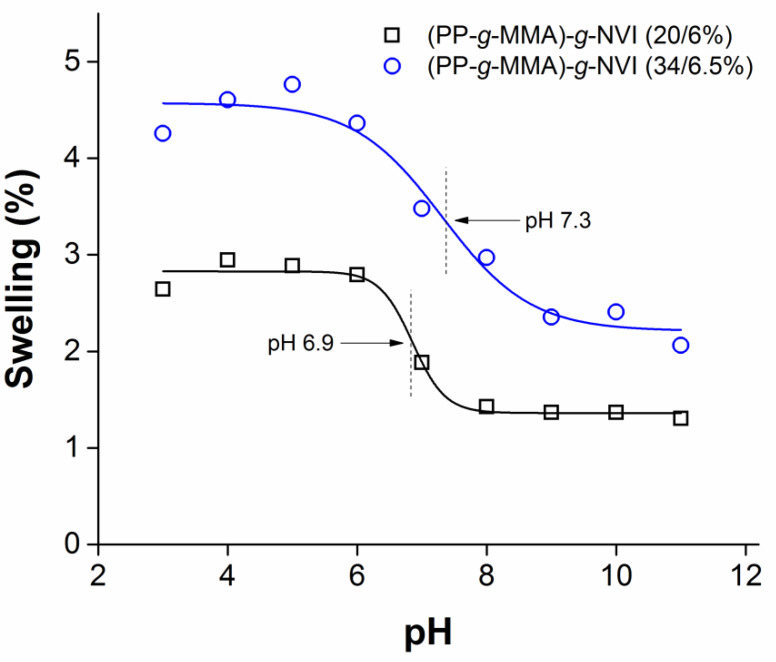
Critical pH, swelling in phosphate buffers.

**Figure 9 ijms-23-00304-f009:**
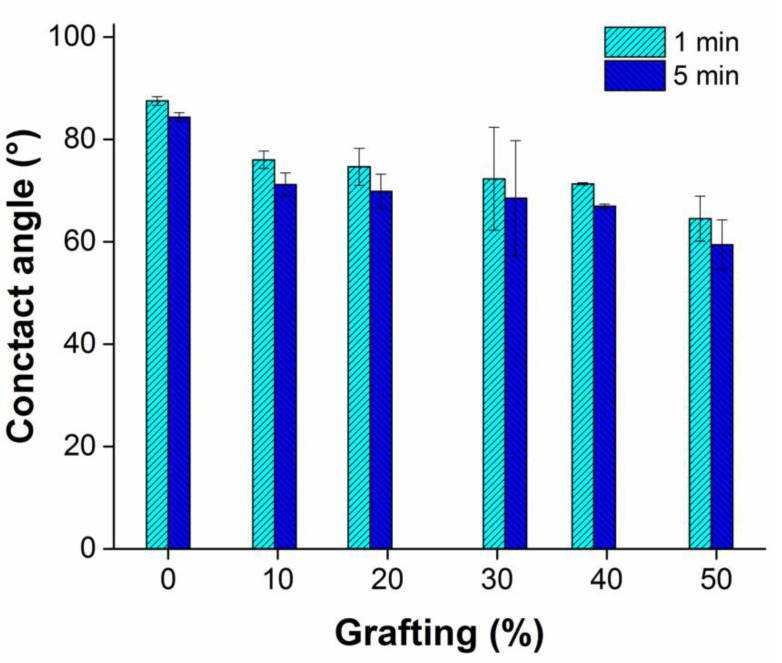
The contact angle of PP-*g*-MMA films showed a change from hydrophobic to hydrophilic.

**Figure 10 ijms-23-00304-f010:**
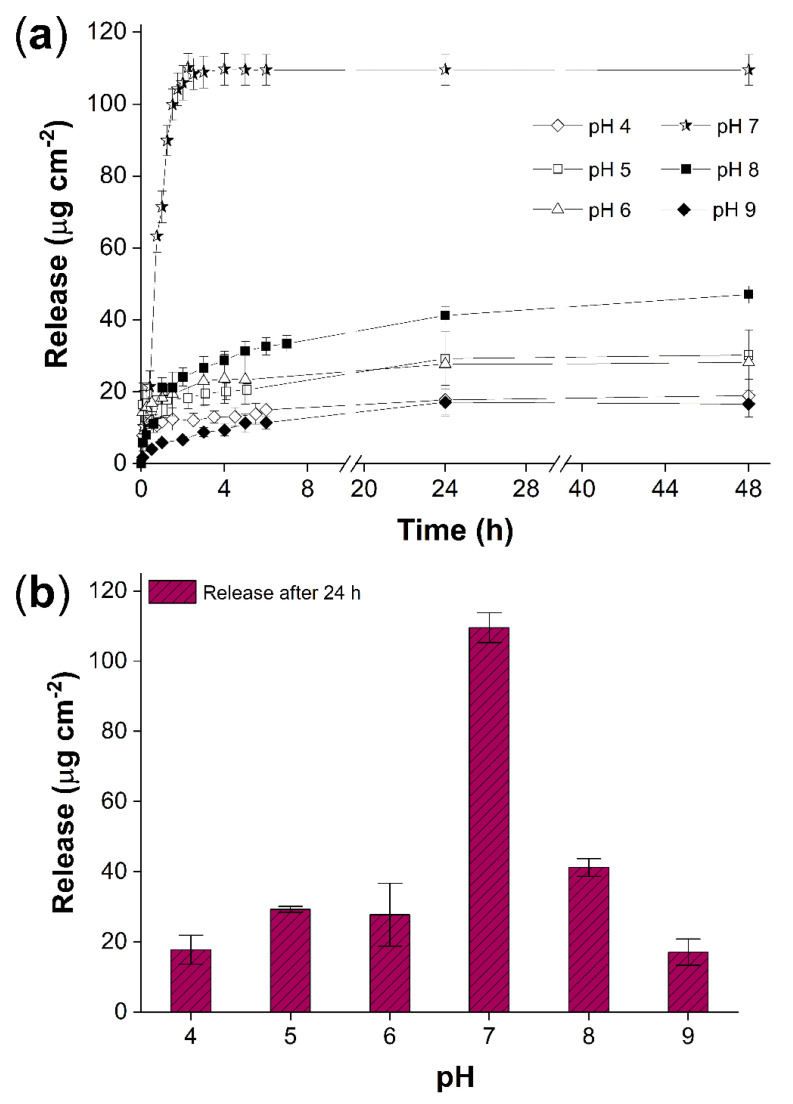
(PP-*g*-MMA)-*g*-NVI (23/5.5%): (**a**) Release of vancomycin in phosphate buffers 0.1 M at different pH and (**b**) release reached at 24 h.

**Table 1 ijms-23-00304-t001:** XPS results of pristine PP, PP-*g*-MMA (17%), and (PP-*g*-MMA)-*g*-NVI (77.5/5%): elemental composition used atomic sensitivity factor of C 1 s: 0.314, O 1 s: 0.733 and N 1 s: 0.499.

Film	Atomic (%)		
	C	O	N
PP	100	-	-
PP-*g*-MMA (17%)	75.28	24.51	-
(PP-*g*-MMA)-*g*-NVI (77.5/5%)	72.59	22.90	3.76

**Table 2 ijms-23-00304-t002:** Results of TGA (weight loss, decomposition temperature, residue) analyses of pristine non-irradiated PP film and films after single and binary grafting.

Film	10 wt% Loss (°C)	Td (°C)	Residue 800 °C (%)
PP	411.7	458.5	8.30
PP-*g*-MMA (25%)	358.5	370.3, 453.6	4.1
(PP-*g*-MMA)-*g*-NVI (19.5/6%)	303.4	291.32, 406.9, 455.9	2.3

**Table 3 ijms-23-00304-t003:** Reaction conditions of grafting polymerization of MMA by pre-irradiation oxidative and grafting degree.

Experiment	Dose (kGy)	Time (h)	Temperature (°C)	Concentration (vol%)	Grafting (%)
Dose	5	16	65	20	10
Dose	10	16	65	20	18
Dose	15	16	65	20	31
Dose	20	16	65	20	39
Dose	25	16	65	20	49.5
Time	5	5	65	30	6
Time	5	10	65	30	12
Time	5	16	65	30	18
Time	5	24	65	30	23
Time	5	26	65	30	31
Concentration	15	16	70	20	8.5
Concentration	15	16	70	40	19.5
Concentration	15	16	70	60	34
Concentration	15	16	70	80	29.5
Concentration	15	16	70	100	77.5
Temperature	5	16	60	30	15.5
Temperature	5	16	70	30	17.5
Temperature	5	16	80	30	25
Temperature	5	16	90	30	23.5

**Table 4 ijms-23-00304-t004:** (PP-*g*-MMA)-*g*-NVI, reaction conditions of NVI grafting by the direct method.

MMA Grafting (%)	Dose (kGy)	Concentration (vol%)	Total Grafting MMA/NVI (%)
8.5	15	50	8.5/4
19.5	15	50	19.5/6
23	15	50	23/5.5
34	15	50	34/6.5
77.5	15	50	77.5/5

## Data Availability

Not applicable.

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
