# Peer review of "Polypropylene Graft Poly(methyl methacrylate) Graft Poly(N-vinylimidazole) as a Smart Material for pH-Controlled Drug Delivery"

_ijms, 2021, doi:10.3390/ijms23010304_

Round 1

Reviewer 1 Report

The copolymer has been obtained through a versatile two-step route; pristine PP is exposed to gamma rays and grafted using methyl methacrylate (MMA), then N-vinylimidazole (NVI) is grafted onto the copolymer PP-g-MMA by simultaneous irradiation. The characterization including FTIR, SEM, TGA, XPS has been performed. The copolymer (PP-g-MMA)-g-NVI has been loaded with vancomycin, and the drug released has been quantified by UV-vis spectrophotometry at different pH. The surface of (PP-g-MMA)-g-NVI exhibit pH-responsiveness and moderate hydrophilicity, suitable properties for controlled drug release. Some specific comments should be pointed out as following.

  1. The error bars are necessary in Fig. 1, 7 and 8.
  2. What’s the size of this film including length, width and height? The authors need to point out in the revised manuscript.
  3. Various characterization methos have been provided to explore the properties of the film in the manuscript including FTIR, SEM, TGA, XPS and other methods. However, the practical application of drug delivery and release is not enough. Please add the antimicrobial experiment based on the films in the revised manuscript.

Author Response

Comments and Suggestions for Authors

The copolymer has been obtained through a versatile two-step route; pristine PP is exposed to gamma rays and grafted using methyl methacrylate (MMA), then N-vinylimidazole (NVI) is grafted onto the copolymer PP-g-MMA by simultaneous irradiation. The characterization including FTIR, SEM, TGA, XPS has been performed. The copolymer (PP-g-MMA)-g-NVI has been loaded with vancomycin, and the drug released has been quantified by UV-vis spectrophotometry at different pH. The surface of (PP-g-MMA)-g-NVI exhibit pH-responsiveness and moderate hydrophilicity, suitable properties for controlled drug release. Some specific comments should be pointed out as following.

  1. The error bars are necessary in Fig. 1, 7 and 8.

Answer: In those Figures error bars were not added because there were not enough experiments performed.

  1. What’s the size of this film including length, width and height? The authors need to point out in the revised manuscript.

Answer: This information is in section “4.2.1 Grafting polymerization of MMA using the oxidative pre-irradiation method”, PP films of 3x2x0.18 cm (wide, length, and thickness).

  1. Various characterization methos have been provided to explore the properties of the film in the manuscript including FTIR, SEM, TGA, XPS and other methods. However, the practical application of drug delivery and release is not enough. Please add the antimicrobial experiment based on the films in the revised manuscript.

Answer: For this manuscript, we focused to the synthesis and given that vancomycin is a commercial antibiotic with probed activity new references are already included to support it.

Thank you for your review.

Reviewer 2 Report

This paper describes the use of 60Co radiation to modify the surface of polypropylene by grafting poly methyl methacrylate to it and then using it a drug delivery system./

Overall, the work is well-done but there are presentation issues to address. 

Section 4 methods should be before section 2 results in my opinion. Section 3 is not a discussion, it is a conclusion and needs to ne merged with section 5, the conclusions.

Please make sure to incorporate error bars throughout the text as well as in all figures, the latter where possible. The authors do present error bars in figures which show that the values in the text are not accurate to those amounts. This must be fixed before acceptance.

For figure 1, give the linear fits where possible. Add error bars to figure 1 as well.

Reference 24 is not useful as it not readily accessible.  Please give a reference to a more accessible journal. This data has been around for a long time.

I really doubt that they can measure the atomic percent to so many decimal places using XPS.

Please give more references for the IR and XPS assignments.

On line 138, please explain the use of the slash for 19.5/6%. Thy use this nomenclature throughout the manuscript and I have no idea what it means.

There is no way that they have temperatures measured to two decimal place in degrees C. Please fix this.

Error bars are needed in figure 7.

In figure 8, how accurate are the pH values?

Error bars are missing for 40% grafting in figure 9.

The results in figure 9 show that the degrees for the wetting angle are not good to one degree. Give the error bars ion the text as well.

The release rate accuracies near line 210 are not good to two decimal places.

Tables 3 and 4 need error bars as appropriate and explanations for the MMA/NVI notation.

Overall, the work is OK but the presentation needs work.

Author Response

Comments and Suggestions for Authors

This paper describes the use of 60Co radiation to modify the surface of polypropylene by grafting poly methyl methacrylate to it and then using it a drug delivery system./

Overall, the work is well-done but there are presentation issues to address.

Section 4 methods should be before section 2 results in my opinion. Section 3 is not a discussion, it is a conclusion and needs to ne merged with section 5, the conclusions.

Answer: Thank you very much for the recommendation, but it is no possible because the template for IJMS has this order 1. Introduction, 2. Results, 3. Discussion, 4. Materials and Methods, etc.

Please make sure to incorporate error bars throughout the text as well as in all figures, the latter where possible. The authors do present error bars in figures which show that the values in the text are not accurate to those amounts. This must be fixed before acceptance.

Answer: We already checked error bars and values to graphs of contact angle and release; this information is consistent with the explained in the text.

For figure 1, give the linear fits where possible. Add error bars to figure 1 as well.

Answer: We agree, the fitting has been incorporated, but error bars do not apply to figures 1 and 2.

Reference 24 is not useful as it not readily accessible. Please give a reference to a more accessible journal. This data has been around for a long time.

Answer: New references 24,25, and 26 were added to support the IR analysis.

I really doubt that they can measure the atomic percent to so many decimal places using XPS.

Answer: In XPS is normal to report the atomic% with two decimal places since this technique is very sensitive and accurate.

Please give more references for the IR and XPS assignments.

Answer: New references from 25 to 29 have been added to support the results.

On line 138, please explain the use of the slash for 19.5/6%. Thy use this nomenclature throughout the manuscript and I have no idea what it means.

Answer: At the end of section 2.1, this nomenclature using the slash is explained. Numbers denote the percent of the first and second monomer grafted respectively.

There is no way that they have temperatures measured to two decimal place in degrees C. Please fix this.

Answer: Thank you very much for the observation, we obtained these values from DSC instrument and the additional decimal was removed.

Error bars are needed in figure 7.

Answer: This experiment did not have a statistical study.

In figure 8, how accurate are the pH values?

Answer: Error bars were not included because the objective was to point out the critical pH.

Error bars are missing for 40% grafting in figure 9.

Answer: Error bar is not missed; the range is just narrow.

The results in figure 9 show that the degrees for the wetting angle are not good to one degree. Give the error bars ion the text as well.

Answer: Results in Figure 9 are correct, and errors were added to text.

The release rate accuracies near line 210 are not good to two decimal places.

Answer: It has been corrected, now all are displayed to one decimal place.

Tables 3 and 4 need error bars as appropriate and explanations for the MMA/NVI notation.

Answer: These graft values do not have a statistical study. Regarding an explanation for the notation with the slash this was already added.

Overall, the work is OK but the presentation needs work.

Answer: Thank you very much for your positive comments.